# Changes in Pain-Related Psychological Distress After Surgery in Patients with Musculoskeletal Injury

**DOI:** 10.3390/ijerph22060857

**Published:** 2025-05-30

**Authors:** Grant H. Cabell, Billy I. Kim, Kevin A. Wu, Emily J. Luo, Clark Bulleit, Nicholas J. Morriss, Trevor A. Lentz, Brian C. Lau

**Affiliations:** 1Department of Orthopaedic Surgery, Duke University Medical Center, 3475 Erwin Rd., Durham, NC 27705, USA; grant.cabell@duke.edu (G.H.C.); brian.lau@duke.edu (B.C.L.); 2Department of Orthopaedic Surgery, Hospital for Special Surgery, 535 E 70th St, New York, NY 10021, USA; kimbil@hss.edu; 3School of Medicine, Duke University, Durham, NC 27710, USA; kevin.a.wu@duke.edu (K.A.W.); emily.luo@duke.edu (E.J.L.); clark.bulleit@duke.edu (C.B.); 4Department of Orthopaedic Surgery, University of Rochester Medical Center, 10 Miracle Drive, Rochester, NY 14623, USA; nicholasjmorriss@gmail.com; 5Duke Clinical Research Institute, Duke University, 3475 Erwin Rd., Durham, NC 27705, USA

**Keywords:** psychological distress, surgery, pain, resilience

## Abstract

(1) Background: Pain experiences are shaped by both physical injury and psychological distress, posing challenges for orthopedic care. While surgery may alleviate injury-related distress, the extent of psychological improvement post surgery remains unclear. Thus, the purpose of this study was to evaluate changes in general and pain-specific psychological distress after surgical intervention for musculoskeletal injury. (2) Methods: A retrospective review was conducted on 133 patients who underwent musculoskeletal surgery between February 2020 and August 2022 by a single sports medicine fellowship-trained surgeon. Psychological distress was assessed using the Optimal Screening for Prediction of Referral and Outcome Yellow-Flag (OSPRO-YF) tool, both before and at least six months after surgery. Pre- and postoperative scores were compared using paired *t*-tests, and clinically meaningful changes were evaluated using a distribution-based minimal clinically important difference (MCID) threshold. (3) Results: Significant reductions were found in total OSPRO-YF scores and several subdomains including fear avoidance (physical activity and work), kinesiophobia, and pain anxiety. However, 88% of patients showed no meaningful change in overall psychological distress. In patients with high baseline distress, over 20% showed meaningful improvement in six subdomains. (4) Conclusions: Psychological distress often persists after musculoskeletal surgery. Targeted psychological interventions may benefit patients with high preoperative distress.

## 1. Introduction

Pain is a complex, multifaceted phenomenon that can be difficult to assess and quantify due to its subjective nature [1,2]. In orthopedic sports medicine settings, pain presents a unique treatment challenge not only due to the physical injury causing the pain, but also due to the associated psychological effects of injury and surgery. General and pain-specific psychological distress is common among athletes at all levels with musculoskeletal injury [3]. Psychological interventions can be effective in addressing psychological distress, such as cognitive–behavioral techniques to reduce the fear of re-injury and improve psychological coping [4]. Appropriate treatment is predicated on the use of a valid and comprehensive assessment of general and pain-specific psychological distress to identify those at risk of persistent pain or poor quality of life [5,6]. The 10-item Optimal Screening for Prediction for Referral and Outcome Yellow-Flag (OSPRO-YF) assessment tool efficiently measures various psychological constructs to provide orthopedic clinicians with insight into a patients’ risk for persistent pain and pain-related disability [7,8].

Pain intensity and disability are associated with various general and pain-specific psychological distress characteristics in patients with musculoskeletal complaints [7,9,10,11,12]. However, most studies that evaluate psychological distress in surgical populations are cross-sectional or designed to predict outcomes using baseline or one-time measures [13,14,15,16]. Few studies are specifically designed to assess psychological distress changes over time [17,18]. Psychological distress in the setting of musculoskeletal injury has been associated with prolonged disability, pain, and negative postoperative outcomes [19], Therefore, it is crucial that there is a more comprehensive understanding of distress in order to improve surgical care. Understanding if and how distress changes following surgery will provide insight into which characteristics are most likely to be modifiable versus more stable psychological traits. In turn, this information would help guide decisions about the timing and selection of psychological interventions before or after surgery. An important impact of distress characteristics like fear avoidance, pain catastrophizing, and low self-efficacy is that they can prompt unhelpful behaviors like poor adherence to rehabilitation activities (especially if painful) and the avoidance of physical activity that promotes functional recovery. Understanding the stability of these distress characteristics or the extent to which they change has important clinical implications. For example, characteristics that show minimal change might be considered traits that are important to consider in establishing prognosis following surgery or even for determining whether patients are appropriate candidates for surgery versus non-surgical treatment. Characteristics that show greater potential for change may be important treatment targets. These characteristics may also be more important to monitor over time, which has practical benefits of streamlining psychological assessment following surgery. Modifiable characteristics could also be considered in goal-setting and for setting post-surgical expectations.

Thus, the primary aim of this study was to evaluate changes in general and pain-specific psychological distress following surgical intervention for musculoskeletal injury. We hypothesized that most general and pain-specific psychological distress characteristics are innate traits, and thus we will see minimal change comparing pre- and postoperative measurements.

## 2. Materials and Methods

### 2.1. Study Setting and Cohort

This was a retrospective study conducted at a tertiary-care academic medical center. Once Institutional Review Board approval was obtained, the institutional electronic medical record (EMR) was used to identify consecutive patients who underwent surgical treatment for a musculoskeletal injury from one sports medicine fellowship-trained orthopedic surgeon between February 2020 and August 2022. This time period reflects the beginning of routine OSPRO-YF collection in the clinic and allowed for the adequate collection of postoperative data. For the preoperative measure, we collected scores from the OSPRO-YF completed during the patient’s closest appointment to surgery. For the postoperative measure, we collected OSPRO-YF scores for at least 6 months after surgery. The OSPRO-YF completed after but closest to the 6-month time point was selected as the postoperative measure. We only included patients with complete preoperative and postoperative OSPRO-YF scores. Those without complete data (patients lost to follow up or who decided not to complete the survey) were excluded from this analysis. Patient demographics including age, race, smoking status, BMI, and American Society of Anesthesiologists (ASA) classification were obtained from the EMR. We also collected information on other procedures performed during surgery in addition to the surgery performed for the primary condition. This could include, for example, performing a meniscal repair alongside anterior cruciate ligament reconstruction, the latter of which is the primary surgical procedure. This variable could be considered a proxy for the intensity or complexity of the overall surgical procedure.

### 2.2. Study Tools

The OSPRO-YF is a self-report tool that provides practitioners with a comprehensive assessment of general and pain-specific psychological distress without overburdening the patient [7,8]. The OSPRO-YF estimates what patients would score on common legacy questionnaires measuring the following general mood-specific psychological constructs: Depression (Patient Health Questionnaire, PHQ-9), Trait Anxiety (State-Trait Anxiety Inventory, STAI), and Trait Anger (State-Trait Anger Expression Inventory, STAXI). It also assesses the following psychological constructs that are pain-specific: Fear Avoidance for Work (Fear-Avoidance Beliefs Questionnaire Work Subscale, FABQ-W), Fear Avoidance for Physical Activity (Fear Avoidance Beliefs Questionnaire Physical Activity Subscale, FABQ-PA), Pain Catastrophizing (Pain Catastrophizing Scale, PCS), Kinesophobia (Tampa Scale of Kinesophobia, TSK-11), Pain Anxiety (Pain Anxiety Symptoms Scale, PASS), Pain Self-Efficacy (Pain Self-Efficacy Questionnaire, PSEQ), Self-Efficacy for Rehabilitation (Self-Efficacy for Rehabilitation, SER), and Pain Acceptance (Chronic Pain Acceptance Questionnaire, CPAQ). In addition to legacy measure score estimates, the tool also provides an “OSPRO-YF Total Score”, which is a composite measure of overall pain-related psychological distress where higher scores indicate higher overall distress. This measure has been validated as predictive of patient-reported clinical outcomes for pain and disability for a variety of musculoskeletal conditions [9,20,21]. The 10-item version of the tool was used in this study; an example is seen in Table 1. Test–retest reliability of the tool has been reported as good to excellent [12].

### 2.3. Statistical Analysis

Continuous variables are presented as means and standard deviations, while categorical variables are displayed as counts and percentages. Using previously established methods, OSPRO-YF data were used to calculate scores for each of the 11 legacy psychological questionnaires [7]. Paired *t*-tests were then used to compare preoperative and postoperative scores for each of the 11 score estimates. A distribution-based minimal clinically important difference (MCID) was calculated for each of the 11 constructs using the standard deviation of the change scores and multiplying by 1.5, following similar methods as was performed in [22]. We used this approach as there were no consistent MCID values established for all legacy measures estimated by the OSPRO-YF in a sports medicine population. To determine the proportion of people meeting MCID thresholds for each construct, individual change scores from preoperation to postoperation for each questionnaire and total OSPRO-YF score were then calculated and compared to the MCID. In an exploratory analysis, we selected patients with high preoperative levels of general and pain-specific psychological distress (i.e., highest quartile of individual questionnaire score estimates for negative psychological constructs, lowest quartile of individual questionnaire score estimates for positive psychological constructs) and determined the proportion meeting MCID postoperatively. Statistical analyses were performed using Rv3.6.1 (RFoundation, Vienna, Austria). Findings were considered statistically significant if *p* < 0.05.

## 3. Results

Our study population included 133 patients. Patients had a mean age of 49.0 +/− 17.2 years old, were predominantly White/Caucasian (70.7%), and were predominantly nonsmokers (96.2%). The cohort was made up of 44.4% of males. The most common joints operated on were the knee (72 patients, 54.1%) and the shoulder (45 patients, 33.8%), though procedures were also performed on the foot/toes, ankle, hip/pelvis, and humerus/elbow (Table 1). Knee arthroscopy was the most common form of operation (45.9%). Of the 133 patients, 73 (54.9%) patients had additional procedures performed. The full distribution of demographics is shown in Table 2.

Mean OSPRO-YF Total Score (20.0 preoperatively, 18.5 postoperatively, *p* = 0.027) and the individual constructs Fear Avoidance for Physical Activity (16.4, 13.5, <0.001), Fear Avoidance for Work (16.3, 13.1, 0.002), Kinesophobia (24.7, 22.8, 0.001), and Pain Anxiety (30.8, 27.8, 0.027) all decreased significantly after surgery. No other changes met our threshold for statistical significance (Table 3) (Figure 1).

When assessing meaningful change using a distribution-based MCID, 88% of patients had no meaningful change in the OSPRO-YF Total Score after surgery (Figure 2). When assessing individual questionnaire score changes, 10 out of 11 constructs had 80% or more patients show no meaningful change (Table 4). Only one construct, Fear Avoidance for Physical Activity, had 20% or more patients with a meaningful improvement (30 patients, 22.6%) in questionnaire score estimates postoperatively.

### Exploratory Analysis of Those with High Preoperative Distress

When considering only those with high levels of preoperative distress (i.e., score in highest population quartile for negative distress factors and lowest population quartile for positive/adaptive distress factors like self-efficacy), 6 of 11 constructs had 20% or more patients show meaningful improvement. Fear Avoidance for Physical Activity had the greatest proportion of patients show a meaningful decrease after surgery (26 patients, 34.7%), followed by Pain Catastrophizing (17 patients, 32.1%), Depression (10 patients, 27.8%), Pain Anxiety (14 patients, 22.2%), Kinesiophobia (17 patients, 21.0%), and Fear Avoidance for Work (15 patients, 20.5%) (Figure 3).

## 4. Discussion

Understanding psychological distress changes following surgical treatment can help musculoskeletal clinicians provide optimal care for their athletes [23,24,25]. This analysis showed that overall OSPRO-YF Total Score and numerous pain-specific measures of psychological distress had statistically significant improvements following surgery. However, the clinical meaningfulness of these changes using distribution-based MCIDs was negligible to modest for most patients, even among those with high levels of distress preoperatively. It should be noted that we observed significant variability in the overall OSPRO-YF score change at the patient level, even if average changes were modest. These findings suggest the need to better understand common trajectories of psychological distress present after surgery for musculoskeletal injury. Patients with a high number of preoperative yellow flags or who show markers of a worsening pain-related psychological distress trajectory in the postoperative stage may be at risk of having prolonged recovery or worse functional outcomes [26]. These patients would likely benefit from rehabilitation that includes a psychological intervention in addition to standard biomedical treatment [27]. Furthermore, patients who have low levels of psychological distress may progress through rehabilitation protocols at an expedited rate as their recovery will be more heavily dependent on physiological healing and recovery rather than psychological determinants.

In this analysis, we observed the most meaningful changes in fear-avoidance beliefs after surgical treatment. Intuitively, this may be expected. As a patient becomes more comfortable and confident in the body after surgery, clinicians may expect fear avoidance cognitions and behaviors to subside. Our findings align with those of Domenech et al. who found that kinesophobia decreased significantly after treatment of chronic anterior knee pain [28]. Chmielewski and George had similar findings in patients after ACL reconstruction, finding that kinesophobia decreased significantly over time at both 4 and 12 weeks postoperatively [29].

Our findings are in contrast with surgical studies in non-musculoskeletal populations that have found psychological distress changes to be highly variable following surgery. For example, Hellstadius et al. studied anxiety and depression prevalence in patients who underwent esophageal cancer surgery, and found that anxiety remained stable over time, while depression increased from pre-surgery to six months postoperatively [30]. In prostate cancer, Occhipinti et al. found that the prevalence of high psychological distress decreased in the first two years after surgery [31]. Further research is needed to investigate if there are inherent differences in general or pain-related distress trajectories with musculoskeletal injury, or if there are factors in the recovery process after all types of surgery that influence the stability of psychological traits more universally.

The findings of this study have important clinical implications. Operative treatment alone may not be enough to fully remedy patients’ psychological distress associated with injury. Persistent distress is likely to result in decreased function and an inability to engage in desired activities. Although we did not track the content of rehabilitation for these patients, it is unlikely that many received interventions that directly addressed psychological distress. The addition of psychologically focused treatments to rehabilitation (e.g., psychologically informed physical therapy) could be beneficial to patients who demonstrate high levels of distress before and after surgery. Interestingly, we did not observe improvements in so-called adaptive or protective factors like pain acceptance or self-efficacy. These factors are not simply the inverse of unhelpful pain coping, but reflect unique cognitions, skills, and behaviors that patients can be taught to help improve recovery. Finally, our results reinforce the potential benefits of monitoring pain-specific psychological distress. Monitoring distress throughout rehabilitation can help guide the selection and timing of interventions to ensure distress is being appropriately managed.

Clinicians should be cautious in how they use psychological distress assessment results to inform surgical decision making. High levels of psychological distress alone should not bar patients from surgical treatment if they are otherwise indicated for a procedure. Surgery can have many indirect benefits to psychological health (e.g., reduced pain, improved mobility) and those with high levels of distress should be exposed to a rehabilitation protocol focused on addressing distress. However, patients with high psychological distress and borderline pathology where either operative or nonoperative care is reasonable may benefit from trying nonoperative treatment rather than an invasive procedure as an initial course of care. For example, a patient with moderate knee osteoarthritis and high levels of psychological distress may have a greater improvement in their quality of life from nonoperative treatment consisting of glucocorticoid injections, physical therapy, behavioral health therapy, and lifestyle modifications than an invasive procedure [32]. Other factors such as social support, psychological counseling, changes in employment/housing, and other social needs should also be considered during pre-surgical evaluation and optimization of postoperative care. These factors were not measured in this study but could influence the trajectory of psychological distress before and after surgery. Further research is needed to fully examine how to best integrate psychological optimization around the surgical encounter.

### Limitations

There were several limitations that should be considered with this study. Despite the validity of the OSPRO-YF, the use of this and other psychological assessment tools assumes that patients are completing questions truthfully and appropriately. Second, the OSPRO-YF is designed as a screening tool that provides estimates of full-length legacy questionnaire scores. There is an inherent measurement error in those score estimates which can affect the precision of change score estimates. Third, scores were assessed at only two time points and thus may not accurately reflect the day-to-day variability in distress over the course of a 6-month recovery. Moreover, while our policy is to collect all scores at 6 months following surgery, not all patients completed their measures at exactly 6 months, consistent with the natural variability in real-world clinical follow-up. Further, baseline data collection occurred during the patient evaluation prior to surgery and this date could have been a few days to a few weeks prior to surgery depending on the availability of surgical slots, the urgency of the procedure/condition, and patient availability. Fourth, since validated MCIDs are not available for all 11 measures estimated by the OSPRO-YF, we decided to use a consistent distribution-based meaningfulness threshold for each measure. This approach benefits from having a similar threshold across constructs but may not adequately reflect meaningful change to all patients. Thus, future work should incorporate anchor-based thresholds that are more patient-centered and may more accurately reflect meaningful change from the patient perspective. As described previously, factors like exposure to counseling and the presence of social needs were not measured in this study but could influence meaningful changes in distress over time. Finally, all patients were recruited from one surgeon’s clinic at one center. Thus, our findings may have limited generalizability and results would need to be replicated in larger samples across more diverse settings.

## 5. Conclusions

Most general and pain-related psychological distress characteristics do not meaningfully change from the pre- to postoperative time period in patients undergoing surgery for sports-related injury. Measures of depression, kinesophobia, and fear-avoidance beliefs were most likely to change in patients with high distress preoperatively and may be important targets for psychologically focused treatments. Further research is needed to investigate longer-term psychological changes and how these changes impact outcomes.

## Figures and Tables

**Figure 1 ijerph-22-00857-f001:**
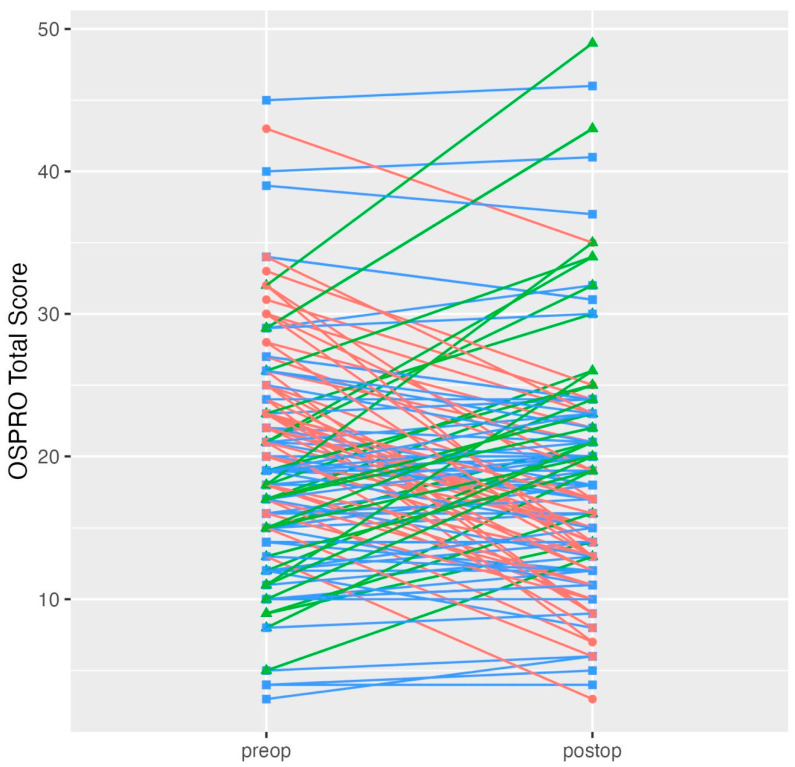
Individual OSPRO Total Score changes preoperatively to postoperatively. Degrees of freedom was 132 for all paired tests. Note: Scores in red decreased significantly, those in green increased significantly, and those in blue showed no significant change. Abbreviations: OSPRO = Optimal Screening for Prediction of Referral and Outcome.

**Figure 2 ijerph-22-00857-f002:**
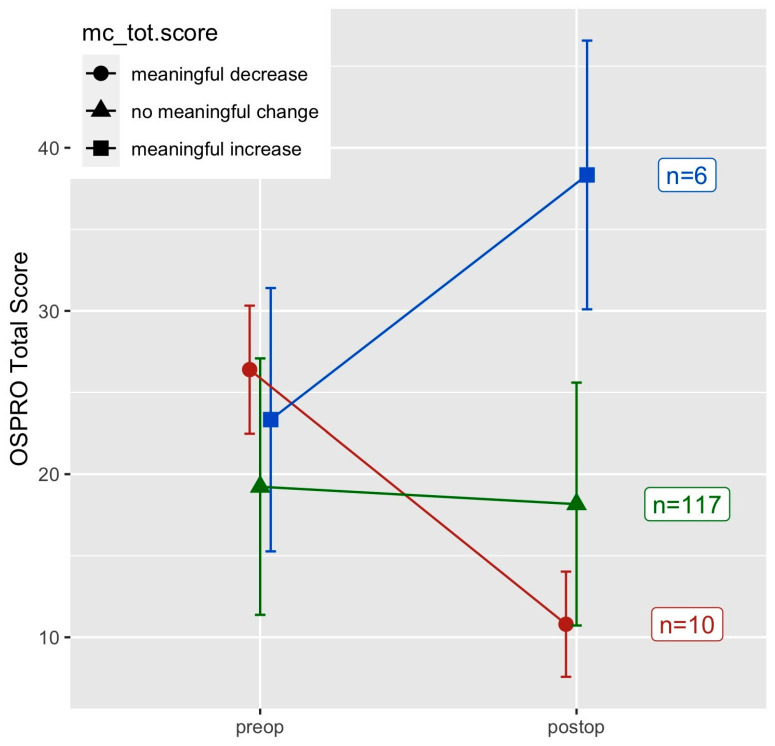
MCID changes in OSPRO-YF Total Score.

**Figure 3 ijerph-22-00857-f003:**
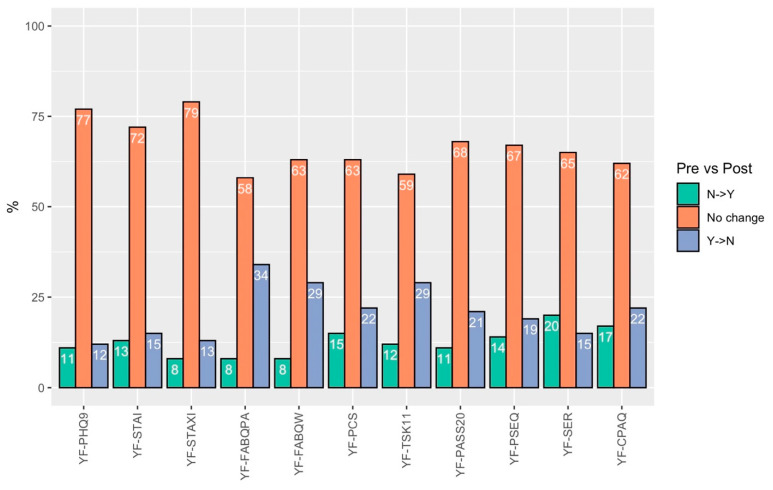
Percentage of patients with changes in yellow flag status among those with high levels of preoperative distress.

**Table 1 ijerph-22-00857-t001:** Example items of the 10-item Optimal Screening for Prediction for Referral and Outcome Yellow-Flag (OSPRO-YF) tool.

Psychological Domain	Example Item (Paraphrased)	Response Range
Negative Mood	“I am hot-headed”	1–4 (Almost never–Almost always)
Poor Coping	“I can’t stop thinking about my pain” “I refrain from activities that may aggravate my pain”	0–4 (Not at all–All the time) 0–6 (Completely disagree–Completely agree)
Positive Affect	“I live a full life despite chronic pain” “My emotional state does not deter me from performing therapy”	0–4 (Never true–Always true) 0–10 (I can’t do it–I can do it)

Adapted OSPRO-YF tool from Lentz et al. 2016 [7] (https://www.jospt.org/doi/10.2519/jospt.2016.6487 accessed on 2 March 2025).

**Table 2 ijerph-22-00857-t002:** Patient characteristics.

	Overall
*N*	133
Age in years, mean (sd)	49.0 (17.2)
Gender, *n* (%)	
Male	59 (44.4)
Female	74 (56.6)
Race, *n* (%)	
White	94 (70.7)
Black/AA	28 (21.1)
Mixed/Other/Not Reported	11 (8.3)
ASA, mean (sd)	2.19 (0.62)
BMI, mean (sd)	29.91 (6.71)
Current Smoker, *n* (%)	5 (3.8)
Laterality = Right, *n* (%)	61 (45.9)
Anatomic Location of Surgery, *n* (%)	
Femur/Knee	72 (54.1)
Foot/Toes	1 (0.8)
Humerus/Elbow	2 (1.5)
Leg/Ankle	8 (6.0)
Other MSK	4 (3.0)
Pelvis/Hip	1 (0.8)
Shoulder	45 (33.8)
CPT Code Category	
Femur/Knee Arthroscopy	61 (45.9)
Femur/Knee Excision	1 (0.8)
Femur/Knee Repair/Revision/Reconstruction	10 (7.5)
Foot/Toes Fracture and/or Dislocation	1 (0.8)
Humerus/Elbow Repair/Revision/Reconstruction	2 (1.5)
Leg/Ankle Arthroscopy	2 (1.5)
Leg/Ankle Fracture and/or Dislocation	2 (1.5)
Leg/Ankle Incision	1 (0.8)
Leg/Ankle Repair/Revision/Reconstruction	3 (2.3)
Other MSK Intro or Removal	4 (3.0)
Pelvis/Hip Fracture and/or Dislocation	1 (0.8)
Shoulder Arthroscopy	42 (31.6)
Shoulder Fracture and/or Dislocation	2 (1.5)
Shoulder Repair/Revision/Reconstruction	1 (0.8)

Abbreviations: sd = standard deviation; AA = African American; MSK = musculoskeletal; CPT = Current Procedural Terminology.

**Table 3 ijerph-22-00857-t003:** Paired comparisons of preoperative and postoperative OSPRO score and questionnaire estimates.

	Preop	Postop	Mean Difference	T-Statistic	*p*-Value *
*N*	133	133			
OSPRO Total Score, mean (sd)	20.0 (7.9)	18.5 (8.6)	−1.44	−2.24	**0.027**
PHQ9, mean (sd)	5.2 (4.1)	4.8 (4.6)	−0.33	−0.86	0.390
STAI, mean (sd)	36.1 (7.9)	35.7 (8.6)	−0.46	−0.73	0.466
STAXI, mean (sd)	15.2 (3.3)	15.0 (3.7)	−0.29	−1.27	0.207
FABQPA, mean (sd)	16.4 (4.8)	13.5 (5.0)	−2.89	−5.33	**<0.001**
FABQW, mean (sd)	16.3 (11.5)	13.1 (10.7)	−3.22	−3.09	**0.002**
PCS, mean (sd)	16.1 (9.5)	14.7 (10.1)	−1.40	−1.52	0.132
TSK11, mean (sd)	24.7 (5.8)	22.8 (6.0)	−1.88	−3.25	**0.001**
PASS20, mean (sd)	30.8 (14.9)	27.8 (15.8)	−3.00	−2.24	**0.027**
PSEQ, mean (sd)	35.6 (10.6)	36.8 (11.5)	1.12	1.21	0.230
SER, mean (sd)	88.9 (19.4)	87.1 (21.4)	−1.86	−0.99	0.324
CPAQ, mean (sd)	63.3 (15.5)	65.0 (16.3)	1.66	1.26	0.208

* Bolded values indicate *p* < 0.05.

**Table 4 ijerph-22-00857-t004:** Meaningful change in questionnaire score estimates.

	Meaningful Decrease	No Meaningful Change	Meaningful Increase
**Meaningful Change (1.5 × SD) in Questionnaire Score Estimates**			
OSPRO Total Score	10 (7.5)	117 (88.0)	6 (4.5)
PHQ9	10 (7.5)	117 (88.0)	6 (4.5)
STAI	6 (4.5)	123 (92.5)	4 (3.0)
STAXI	6 (4.5)	122 (91.7)	5 (3.8)
FABQPA	30 (22.6)	95 (71.4)	8 (6.0)
FABQW	15 (11.3)	112 (84.2)	6 (4.5)
PCS	17 (12.8)	107 (80.5)	9 (6.8)
TSK11	17 (12.8)	108 (81.2)	8 (6.0)
PASS20	15 (11.3)	111 (83.5)	7 (5.3)
PSEQ	4 (3.0)	117 (88.0)	12 (9.0)
SER	10 (7.5)	113 (85.0)	10 (7.5)
CPAQ	5 (3.8)	116 (87.2)	12 (9.0)
**Meaningful change (1.5 × SD) in patients presenting with YF for each questionnaire**			
PHQ9	10 (27.8)	25 (69.4)	1 (2.8)
STAI	6 (14.6)	34 (82.9)	1 (2.4)
STAXI	5 (12.2)	36 (87.8)	0 (0.0)
FABQPA	26 (34.7)	49 (65.3)	0 (0.0)
FABQW	15 (20.5)	54 (74.0)	4 (5.5)
PCS	17 (32.1)	32 (60.4)	4 (7.5)
TSK11	17 (21.0)	61 (75.3)	3 (3.7)
PASS20	14 (22.2)	46 (73.0)	3 (4.8)
PSEQ	2 (2.4)	68 (82.9)	12 (14.6)
SER	4 (5.6)	58 (80.6)	10 (13.9)
CPAQ	4 (5.6)	56 (77.8)	12 (16.7)

## Data Availability

The data presented in this study are available on request from the corresponding author due to the privacy of patient data.

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
