# Peer review of "Changes in Pain-Related Psychological Distress After Surgery in Patients with Musculoskeletal Injury"

_ijerph, 2025, doi:10.3390/ijerph22060857_

Round 1
Reviewer 1 Report
Comments and Suggestions for Authors
The manuscript is presented as a good proposal for publication. It provides a “demystification” of the relationship of pain with the conclusion that there is no significant change between preoperative and postoperative pain in patients undergoing surgery for sports injuries. And, on the other hand, it opens the way for future studies: “Measures of depression, kinesophobia and fear-avoidance beliefs were more likely to change in patients with high preoperative distress”. However, for publication, it is important to clarify:
- Materials and methods:
As far as I have understood, the data collection was performed with the review of the patients' (computerized) Clinical History. If correct:
- Clarify: The agreement between observations??? Intra-observer or inter-observer? Concordance was performed, how it was performed, and the results.
In relation to the selection of subjects:
- What type of sampling was performed and the total population from which n=133 were drawn.
- Line 83/84: For the preoperative measurement, OSPRO-YF scores completed during the patient's appointment closest to surgery were collected: it is important to state and make clear “closest to surgery” (two days before, three days before, 1 month before...?) may be a confounding factor or bias. Perception may change in relation to the variable “time to procedure”.
Author Response
The manuscript is presented as a good proposal for publication. It provides a “demystification” of the relationship of pain with the conclusion that there is no significant change between preoperative and postoperative pain in patients undergoing surgery for sports injuries. And, on the other hand, it opens the way for future studies: “Measures of depression, kinesophobia and fear-avoidance beliefs were more likely to change in patients with high preoperative distress”. However, for publication, it is important to clarify:
Response: Thank you for these positive comments. We have addressed each below and in the manuscript, as appropriate.
Materials and methods:
As far as I have understood, the data collection was performed with the review of the patients' (computerized) Clinical History. If correct:
- Clarify: The agreement between observations??? Intra-observer or inter-observer? Concordance was performed, how it was performed, and the results.
Response:
Thank you for your comment. Because this study involved retrospective analysis of patient-reported questionnaire data extracted from the electronic medical record (EMR), there was no need for intra- or inter-observer agreement assessment. All data were directly retrieved from structured OSPRO-YF assessments completed independently by patients in clinic and automatically recorded into the EMR system. If this comment is specific to the inter-rater reliability of the OSPRO-YF instrument, prior studies have established ICC values of 0.91-0.94 (Razmjou et al., 2021) and 0.85 (Butera et al., 2021). This has been added to the methods section.
In relation to the selection of subjects:
- What type of sampling was performed and the total population from which n=133 were drawn.
Response:
Patients were identified using consecutive sampling from a prospectively maintained database of patients who underwent surgery with a single sports medicine fellowship-trained orthopaedic surgeon between February 2020 and August 2022. From this database, 133 patients were included who had complete preoperative and ≥6-month postoperative OSPRO-YF data. We clarified that these patients were consecutive in the manuscript.
- Line 83/84: For the preoperative measurement, OSPRO-YF scores completed during the patient's appointment closest to surgery were collected: it is important to state and make clear “closest to surgery” (two days before, three days before, 1 month before...?) may be a confounding factor or bias. Perception may change in relation to the variable “time to procedure”.
Response:
As these data are collected during routine care, there is no consistent number of days prior to surgery that we can report. Questionnaire collection occurs during the patient evaluation prior to surgery and this date could be a few days to a few weeks prior to surgery depending on the availability of surgical slots, urgency of the procedure/condition, and patient availability. We agree that perception could change over time in this pre-operative phase and this is one of the inherent limitations of using real-world data versus data collected specifically for use in research. We have clarified this and its implications in the revision.
Reviewer 2 Report
Comments and Suggestions for Authors
Please, quote some sources for the statement "Most studies that evaluate psychological distress in surgical populations are cross-sectional or designed to predict outcomes using baseline measures."
Please, quote some sources for the statement "Few studies are specifically designed to assess psychological distress changes over time."
Please, consider revising linguistically the sentence "Thus, the primary aim of this study was to evaluate changes general and pain-specific psychological distress after surgical intervention for musculoskeletal injury."
You should enrich your theoretical part with some previous and recent findings related to pain and distress in case of Musculoskeletal Injury and surgery in Patients with Musculoskeletal Injury, see for example:
Aaron RV, Rassu FS, Wegener ST, Holley AL, Castillo RC, Osgood GM, Fisher E. Psychological treatments for the management of pain after musculoskeletal injury: A systematic review and meta-analysis. Pain. 2024 Jan 1;165(1):3-17. doi: 10.1097/j.pain.0000000000002991.
Leech JB, MacPherson KL, Klopper M, Shumway J, Salvatori RT, Rhon DI, Young JL. The relationships between pain-associated psychological distress, pain intensity, patient expectations, and physical function in individuals with musculoskeletal pain: A retrospective cohort study. PM R. 2023 Nov;15(11):1371-1381. doi: 10.1002/pmrj.12983.
Psychological Treatment in the Management of Pain following Musculoskeletal Injury
These are some suggestions, but you may quote any other sources that you consider relevant to your topic.
Please, explain why in Table 1 Number of other procedures, n is not a whole number (0.74)
In the text and in Table 2, you report only p-value of t-test. You should also report the value of t-test and the degrees of freedom.
Figure 2 cannot have such a long title. A part of the title should be a Note below Figure 2 (Scores in red decreased significantly, green 157 increased significantly, and blue scores showed no significant change. Abbreviations: OSPRO = Optimal Screening for Prediction 158 of Referral and Outcome. PHQ9 = Patient Health Questionnaire. STAI = State-Trait Anxiety Inventory. STAXI = State-Trait Anger 159 Expression Inventory. FABQPA = Fear Avoidance Beliefs Questionnaire Physical Activity Subscale. FABQW = Fear Avoidance 160 Beliefs Questionnaire Work Subscale. PCS = Pain Catastrophizing Scale. TSK11 = Tampa Scale of Kinesophobia. PASS20 = Pain 161 Anxiety Symptoms Scale. PSEQ = Pain Self-Efficacy Questionnaire. SER = Self-Efficacy for Rehabilitation. CPAQ = Chronic Pain 162 Acceptance Questionnaire.). The reviewer supposes that each line on this figure refers to one patient.
Figure 1 in fact is a table: "Figure 1: Example Items of the 10-item Optimal Screening for Prediction for Referral and Outcome Yellow-Flag 119 (OSPRO-YF) tool" Please, change its label as a table and re-numerate tables and figures.
Figure 3 to which you refer in the text is missing:
"When assessing meaningful change using a distribution based MCID, 88% of patients had no meaningful change in the OSPRO-YF total score after surgery. (Figure 3)"
Table 3 to which you refer in the text is missing.
"When assessing individ-167 ual questionnaire score changes, 10 out of 11 constructs had 80% or more patients show no mean-168 ingful change. (Table 3)"
Figure 4 to which you refer in the text is missing:
"When considering only those with high levels of preoperative distress (i.e., score in highest 174 population quartile for negative distress factors and lowest population quartile for positive/adaptive 175 distress factors like self-efficacy), 6 of 11 constructs had 20% or more patients show meaningful 176 improvement. Fear Avoidance for Physical Activity had the greatest proportion of patients show a 177 meaningful decrease after surgery (26 patients, 34.7%), followed by Pain Catastrophizing (17 pa-178 tients, 32.1%), Depression (10 patients, 27.8%), Pain Anxiety (14 patients, 22.2%), Kinesiophobia 179 (17 patients, 21.0%), and Fear Avoidance for Work (15 patients, 20.5%). (Table 3) (Figure 4)."
Please, add all necessary Tables and Figures to your manuscript to be fully comprehensible and scientifically supported the results.
Author Response
Please, quote some sources for the statement "Most studies that evaluate psychological distress in surgical populations are cross-sectional or designed to predict outcomes using baseline measures."
Response:
Thank you. We have revised the text to include appropriate citations.
Please, quote some sources for the statement "Few studies are specifically designed to assess psychological distress changes over time."
Response:
Thank you. We have revised the text to include appropriate citations.
Please, consider revising linguistically the sentence "Thus, the primary aim of this study was to evaluate changes general and pain-specific psychological distress after surgical intervention for musculoskeletal injury."
Response:
We revised the sentence to: “Thus, the primary aim of this study was to evaluate changes in general and pain-specific psychological distress following surgical intervention for musculoskeletal injury.”
You should enrich your theoretical part with some previous and recent findings related to pain and distress in case of Musculoskeletal Injury and surgery in Patients with Musculoskeletal Injury, see for example:
Aaron RV, Rassu FS, Wegener ST, Holley AL, Castillo RC, Osgood GM, Fisher E. Psychological treatments for the management of pain after musculoskeletal injury: A systematic review and meta-analysis. Pain. 2024 Jan 1;165(1):3-17. doi: 10.1097/j.pain.0000000000002991.
Leech JB, MacPherson KL, Klopper M, Shumway J, Salvatori RT, Rhon DI, Young JL. The relationships between pain-associated psychological distress, pain intensity, patient expectations, and physical function in individuals with musculoskeletal pain: A retrospective cohort study. PM R. 2023 Nov;15(11):1371-1381. doi: 10.1002/pmrj.12983.
Psychological Treatment in the Management of Pain following Musculoskeletal Injury
These are some suggestions, but you may quote any other sources that you consider relevant to your topic.
Response:
Thank you for the suggested articles. We have incorporated these as well as other studies to better highlight the issue regarding pain and distress in MSK injury and surgery.
Please, explain why in Table 1 Number of other procedures, n is not a whole number (0.74)
Response:
This variable represented the mean number of additional procedures each patient underwent during the index operation. However, as this variable was unclear, we removed it from the table and added to the results the percentage of patients who had an additional procedure performed: “Of the 133 patients, 73 (54.9%) patients had additional procedures performed.” We also now provide more detail about the additional procedure variable in the methods. Note this has been renamed Table 2.
In the text and in Table 2, you report only p-value of t-test. You should also report the value of t-test and the degrees of freedom.
Response:
Please see the appended t-statstic values. Degrees of freedom was 132 for all paired tests, now referenced in the table footnote. All patients included in the study completed pre and post operative surveys. Note this has been renamed Table 3
Figure 2 cannot have such a long title. A part of the title should be a Note below Figure 2 (Scores in red decreased significantly, green 157 increased significantly, and blue scores showed no significant change. Abbreviations: OSPRO = Optimal Screening for Prediction 158 of Referral and Outcome. PHQ9 = Patient Health Questionnaire. STAI = State-Trait Anxiety Inventory. STAXI = State-Trait Anger 159 Expression Inventory. FABQPA = Fear Avoidance Beliefs Questionnaire Physical Activity Subscale. FABQW = Fear Avoidance 160 Beliefs Questionnaire Work Subscale. PCS = Pain Catastrophizing Scale. TSK11 = Tampa Scale of Kinesophobia. PASS20 = Pain 161 Anxiety Symptoms Scale. PSEQ = Pain Self-Efficacy Questionnaire. SER = Self-Efficacy for Rehabilitation. CPAQ = Chronic Pain 162 Acceptance Questionnaire.). The reviewer supposes that each line on this figure refers to one patient.
Response:
Thank you. The description of color codes and abbreviations has been moved below the figure as a Note. The figure title was shortened accordingly.
Figure 1 in fact is a table: "Figure 1: Example Items of the 10-item Optimal Screening for Prediction for Referral and Outcome Yellow-Flag 119 (OSPRO-YF) tool" Please, change its label as a table and re-numerate tables and figures.
Response:
We agree. Figure 1 has been re-labeled as Table 2, and subsequent figures and tables have been re-numbered appropriately.
Figure 3 to which you refer in the text is missing:
"When assessing meaningful change using a distribution based MCID, 88% of patients had no meaningful change in the OSPRO-YF total score after surgery. (Figure 3)"
Response:
Apologies for the oversight. Figure 2 (former Figure 3 due to comment above) has been added to illustrate the distribution of meaningful change in OSPRO-YF Total Score.
Table 3 to which you refer in the text is missing.
"When assessing individ-167 ual questionnaire score changes, 10 out of 11 constructs had 80% or more patients show no mean-168 ingful change. (Table 3)"
Response:
Thank you. Table 4 (former Table 3 due to comment above) has been included to summarize proportions of patients achieving MCID across constructs.
Figure 4 to which you refer in the text is missing:
"When considering only those with high levels of preoperative distress (i.e., score in highest 174 population quartile for negative distress factors and lowest population quartile for positive/adaptive 175 distress factors like self-efficacy), 6 of 11 constructs had 20% or more patients show meaningful 176 improvement. Fear Avoidance for Physical Activity had the greatest proportion of patients show a 177 meaningful decrease after surgery (26 patients, 34.7%), followed by Pain Catastrophizing (17 pa-178 tients, 32.1%), Depression (10 patients, 27.8%), Pain Anxiety (14 patients, 22.2%), Kinesiophobia 179 (17 patients, 21.0%), and Fear Avoidance for Work (15 patients, 20.5%). (Table 3) (Figure 4)."
Response:
We have now included figure (Figure 3 in the revision) to illustrate the subset of patients with high preoperative distress who demonstrated meaningful improvement in psychological measures.
Please, add all necessary Tables and Figures to your manuscript to be fully comprehensible and scientifically supported the results.
Response:
Done
Reviewer 3 Report
Comments and Suggestions for Authors
Very interesting research project. I propose moving tab.1 to results. It is worth adding sections on study limitations and impact on practice. It is worth paying attention to barriers from the staff side in the discussion, e.g. I found an excellent article by Mędrzycka-Dąbrowska W, Dąbrowski S, Gutysz-Wojnicka A, Basiński A. Polish nurses' perceived barriers in using evidence-based practice in pain management. Int Nurs Rev. 2016 Sep;63(3):316-27. doi: 10.1111/inr.12255. and Rababa M, Al-Sabbah S, Hayajneh AA. Nurses' Perceived Barriers to and Facilitators of Pain Assessment and Management in Critical Care Patients: A Systematic Review. J Pain Res. 2021 Nov 5;14:3475-3491. doi: 10.2147/
Comments on the Quality of English Languagelack of competence to assess the quality of language
Author Response
Thank you for these suggestions. We have moved Table 1 but elected not to include the referenced paper as it focuses on critical care which is a substantially different population than that studied in this paper.
Reviewer 4 Report
Comments and Suggestions for Authors
Dear Authors,
The topic addressed in the manuscript is very relevant to both clinical orthopaedic practice and the broader field of health psychology. The use of the OSPRO-YF tool, which has been validated for the assessment of pain-related psychological distress, enhances the relevance and applicability of the findings.
The introduction presents a relevant and timely clinical problem, persistent psychological distress following surgery for musculoskeletal injuries, and clearly outlines the rationale for conducting the study.. Key concepts such as fear avoidance, kinesiophobia, and pain anxiety are introduced in a logical sequence, and the use of the OSPRO-YF tool is well justified. However, the theoretical background could benefit from a little bit of expansion. Specifically, while previous studies are cited to illustrate gaps in understanding psychological distress trajectories after surgery, a more critical synthesis of prior research would help clarify how this study builds upon and differs from existing literature. For instance, it would be useful to elaborate more explicitly on the limitations of cross-sectional designs in prior studies and how the longitudinal aspect of the present study provides added value. Additionally, a very short paragraph on how psychological distress may influence rehabilitation adherence and functional recovery would put more focus on the study’s clinical importance.
The retrospective cohort design is methodologically sound and described in appropriate detail.
Statistical methods are correctly applied. The use of paired t-tests and minimal clinically important difference (MCID) thresholds is suitable for evaluating changes in psychological constructs over time. Still, a more nuanced focus on the limitations of using a distribution-based MCID, particularly in the absence of established reference values for all OSPRO-YF subscales, would give more the analytical component of the study.
The results are presented clearly, both narratively and in tables.
The discussion section is well grounded in existing literature. You can more enriche discussion section by considering additional, unmeasured factors such as social support, psychological counseling, or other interventions that may influence psychological outcomes during recovery. Also, you can clarify what constitutes a “meaningful change” from the patient’s perspective.
The limitations of the study are reported transparently, including concerns related to generalizability, the use of estimated scores from abbreviated tools, and potential variability in follow-up timing.
Best of luck with the publication.
Kind regards.
Author Response
The topic addressed in the manuscript is very relevant to both clinical orthopaedic practice and the broader field of health psychology. The use of the OSPRO-YF tool, which has been validated for the assessment of pain-related psychological distress, enhances the relevance and applicability of the findings.
The introduction presents a relevant and timely clinical problem, persistent psychological distress following surgery for musculoskeletal injuries, and clearly outlines the rationale for conducting the study.. Key concepts such as fear avoidance, kinesiophobia, and pain anxiety are introduced in a logical sequence, and the use of the OSPRO-YF tool is well justified. However, the theoretical background could benefit from a little bit of expansion. Specifically, while previous studies are cited to illustrate gaps in understanding psychological distress trajectories after surgery, a more critical synthesis of prior research would help clarify how this study builds upon and differs from existing literature. For instance, it would be useful to elaborate more explicitly on the limitations of cross-sectional designs in prior studies and how the longitudinal aspect of the present study provides added value. Additionally, a very short paragraph on how psychological distress may influence rehabilitation adherence and functional recovery would put more focus on the study’s clinical importance.
Response:
We expanded the Introduction to better differentiate this longitudinal design from cross-sectional work. Specifically, we now describe how prior designs fail to capture psychological response over time and the clinical relevance of understanding psychological distress. We have also expanded the introduction to include more information about how the findings of this study could benefit clinical practice and decision-making.
The retrospective cohort design is methodologically sound and described in appropriate detail.
Statistical methods are correctly applied. The use of paired t-tests and minimal clinically important difference (MCID) thresholds is suitable for evaluating changes in psychological constructs over time. Still, a more nuanced focus on the limitations of using a distribution-based MCID, particularly in the absence of established reference values for all OSPRO-YF subscales, would give more the analytical component of the study.
Response:
Thank you for this comment. We added in our limitations that that future work should incorporate anchor-based thresholds that are more patient-centered.
The results are presented clearly, both narratively and in tables.
The discussion section is well grounded in existing literature. You can more enriche discussion section by considering additional, unmeasured factors such as social support, psychological counseling, or other interventions that may influence psychological outcomes during recovery. Also, you can clarify what constitutes a “meaningful change” from the patient’s perspective.
Response:
We expanded the discussion to note that unmeasured variables such as psychological counseling, social support, or changes in employment status may contribute to distress and should thus be considered in evaluation. Regarding meaningful change, we discuss in the limitations section that meaningful change in this study was determined using a distribution-based MCID, which may not fully align with individual patient expectations or lived experiences. As above, we clarified that future work should incorporate anchor-based thresholds that are more patient-centered.
The limitations of the study are reported transparently, including concerns related to generalizability, the use of estimated scores from abbreviated tools, and potential variability in follow-up timing.
Round 2
Reviewer 2 Report
Comments and Suggestions for Authors
Thank you for revising your manuscript!